# *Ideonella sakaiensis* Can Metabolize Bisphenol A as a Carbon Source

**DOI:** 10.3390/microorganisms11122891

**Published:** 2023-11-30

**Authors:** Cristian-Emilian Pop, György Deák, Cristina Maria, Gina Ghiță, Alexandru Anton Ivanov, Sergiu Fendrihan, Dan Florin Mihăilescu, Maria Mernea

**Affiliations:** 1Department of Natural and Technological Hazards, National Institute for Research and Development in Environmental Protection, 294 Splaiul Independenței Str., 060031 Bucharest, Romaniaecologos23@yahoo.com (S.F.); 2Department of Anatomy, Animal Physiology and Biophysics, Faculty of Biology, University of Bucharest, 91–95 Splaiul Independenței Str., 050095 Bucharest, Romania; dan.mihailescu@bio.unibuc.ro (D.F.M.); maria.mernea@bio.unibuc.ro (M.M.); 3Non-Governmental Research Organization Biologic, 14 Schitului Str., 032044 Bucharest, Romania; 4Biometric Psychiatric Genetics Research Unit, Alexandru Obregia Psychiatric Hospital, Șoseaua Berceni 10 Str., 041914 Bucharest, Romania

**Keywords:** *Ideonella sakaiensis*, bisphenol A, growth curve, substrate, bioprospecting

## Abstract

Bisphenol A and its analogues represent a significant environmental and public health hazard, particularly affecting the endocrine systems of children and newborns. Due to the growing need for non-pathogenic biodegradation microbial agents as environmentally friendly and cost-effective solutions to eliminate endocrine disruptors, this study aimed to investigate the degradation of bisphenol A by *Ideonella sakaiensis*, based on its currently understood unique enzymatic machinery that is already well known for degrading polyethylene terephthalate. The present study provides novel insights into the metabolic competence and growth particularities of *I. sakaiensis*. The growth of *I. sakaiensis* exposed to bisphenol A exceeded that in the control conditions, starting with 72 h in a 70% nutrient-rich medium and starting with 48 h in a 100% nutrient-rich medium. Computational modeling showed that bisphenol A, as well as its analogue bisphenol S, are possible substrates of PETase and MHETase. The use of bisphenol A as a carbon and energy source through a pure *I. sakaiensis* culture expands the known substrate spectra and the species’ potential as a new candidate for bisphenol A bioremediation processes.

## 1. Introduction

Bisphenol A (2,2-bis(4-hydroxyphenyl) propane; BPA) is the main monomer used in the manufacturing of polycarbonates [1,2]. The compound was produced globally during 2020, totaling over 7 million tons, and future industrial production is forecast to increase by 4% per annum for 2021–2025 [3]. BPA has been used since 1975 as a cross-linking agent and as a pivotal monomeric constituent in the synthesis of epoxy resins. Furthermore, BPA is used in the fabrication of polysulfones and polyetherketones, serving as an antioxidant in certain plasticizers, and as an inhibitory agent in the polymerization process of polyvinyl chloride [4,5]. The compound can pollute the environment either through direct emissions from manufacturing activities (including material recycling processes), or from foundries utilizing BPA in casting sand; it can also indirectly permeate the environment by leaching from waste in landfills [6].

In 1993, Khrishna et al. [7] were the first to report BPA as an endocrine disruptive chemical after accidentally discovering its tendency to leak out of the polymer structure during autoclaving of media held in polycarbonate flasks. Since then, BPA has gained the attention of the scientific community and has since been banned from infants’ plastic products in Canada, Europe, and the USA [8,9]. Although BPA usage has been limited and awareness has been raised, the compound is still largely in use in the plastics industry, and, consequently, it is now considered to be an emergent pollutant with both known and unknown impacts on human health [8,10] and the environment [11,12].

In the development of new strategies for combating plastic waste pollution and plastic-related polluting compounds, microorganisms have gained international attention, especially with the fairly recent discovery of *Ideonella sakaiensis* and its specific enzymatic ability to convert the inert polymer polyethylene terephthalate (PET) into an important major energy and carbon source [13]. Furthermore, the performance, activity, and stability of the enzyme can be improved by mutation (S136E) and can be identified with the use of bioinformatic tools [14]. The metabolic versatility of *I. sakaiensis* was also revealed in a recent study that demonstrated that the species can ferment PET in anaerobic conditions (thus, it was shown to be a facultative anaerobe), and formed a symbiosis with *Geobacter sulfurreducens* in a microbial fuel cell [15]. However, it is currently unknown to what extent, if any, *I. sakaiensis* can be used in a reverse microbial fuel cell by using a redox mediator that facilitates energy transfer to biomass [16] and/or PET degradation. Although *I. sakaiensis* seems to be a metabolically versatile and promising species, it is worth mentioning that it was successfully isolated from 1 out of only 250 samples [13], being one of the very few reports on microbial life with such distinct characteristics [17,18], as bioprospecting efforts continue in this direction.

In the present study, an attempt was made to identify and characterize microorganisms that might present an affinity to BPA, following previous observations on Gram-negative bacteria that survived in highly BPA-contaminated aquatic experimental set ups [19]. Water was sampled from polycarbonate aquaria with natural water (lake water), which contained 10 µg/mL BPA, for 1 week; the 10 µg/mL concentration was chosen based on the 48 h EC50 related to the invertebrate *Daphnia magna* [20]. Several species of bacteria exhibited tolerance to BPA, while others were inhibited; however, *I. sakaiensis* proliferated in the presence of BPA.

To the best of our knowledge, this is the first report on the physiological response of *Ideonella sakaiensis* to BPA exposure.

## 2. Materials and Methods

### 2.1. Microbial Community Analysis

Community-level physiological profiling (CLPP) has been demonstrated to be effective at distinguishing spatial and temporal changes in microbial communities [21]. The CLPP screening was performed using Biolog EcoPlates (Biolog Inc., Hayward, CA, USA), which can be used as both an assay on the stability of a normal microbial population, as well as to detect and assess changes following the introduction of an environmental variable. The 96 wells in the EcoPlate, representing 31 different carbon sources and tetrazolium dye, were inoculated with 150 µL of a water sample sourced from a polycarbonate aquarium, which was used in an experimental aquaculture system exposed to BPA and inhabited by a variety of microorganisms (incubated at 23 °C under controlled circadian conditions: 12 h light/12 h dark, light was provided by a fluorescent 4000 K white light lamp with an intensity of 300 lumens). The plates were incubated at 23 °C, and the metabolic profile formation of the community was recorded photometrically by reading the absorption at 590 nm (BMG Clariostar, Ortenberg, Germany) at 24, 48, 72, and 96 h. The Biolog EcoPlate method does present certain constraints [22], but still, utilizing this method for assessing bacterial CLPP proves to be a more straightforward and cost-effective option compared to DNA or RNA-based profiling. Furthermore, several investigations have indicated that molecular profiling surveys do not exhibit heightened sensitivity when compared to CLPP methods [23,24].

### 2.2. Primary and Final Taxonomic Identification

From the sample, 100 µL was inoculated onto tryptic soy agar (TSA), Drigalski Lactose, Chapman (MSA), nutrient and chromogen agar plates (Mlt-d, Arad, Romania), and spread over the entire surface using a Drigalski spatula and a Petri dish turntable (Schuett-Biotec, Göttingen, Germany); followed by a macroscopic observation of the colonies and their morphology after 24 and 48 h post-incubation at 23 °C.

Final taxonomic identification was performed with a matrix-assisted laser desorption/ionization time-of-flight (MALDI-TOF) mass spectrometer biotyper (MALDI-TOF Microflex LT/SH, Bruker Daltonics, Bremen, Germany). The spots on an MSP 96 target polished steel plate (Bruker Daltonics, Bremen, Germany) were loaded with isolated colonies, and on each spot an α- Cyano-4-hydroxycinnamic acid (Bruker Daltonics, Bremen, Germany) matrix was added along with the stock solution (acetonitrile, trifluoracetic acid, and distilled water). The SR Library Database (Database CD BT BTyp2.0-Sec-Library 1.0) coupled with MBT Compass (v.4.1) software was used for the identification.

### 2.3. Species Exposure to Bisphenol A, Bisphenol S, and 17-ß Estradiol

The identified species, as well as the *Ideonella sakaiensis* 201-F6 strain (DSM 112585, Leibniz Institute, Germany), were grown as pure cultures in nutrient broth (pH 7) and used as an inoculum at similar densities (Abs ≈ 0.500 at OD600, log phase), as all species previously presented with good growth on nutrient agar. From these cultures, 10 µL was used to inoculate tubes of nutrient broth supplemented with 1 mL of aqueous BPA stock solution, to a final concentration of 10µg/mL. Glass vials were used instead of plastic tubes in order to exclude possible interference due to any leached material. The growth of the microorganisms was monitored at OD600 (Implen NP80, München, Germany) for 24 h at 23 °C, this temperature was chosen as it was the temperature of the aquaculture system from which the samples were obtained; further, it is considered to be an ambient room temperature, therefore better reflecting in situ conditions.

Due to the positive growth response by *I. sakaiensis* to BPA, a further investigation was conducted by supplementing nutrient broth media in glass vials (each with a volume of 6.920 mL), with, respectively, 10 µg/mL of BPA, BPS (Bis(4-hydroxyphenyl) sulfone; BPS), as well as 1 µg/mL 17-ß estradiol. All the analytes were prepared in anhydrous ethanol (final volume 70 µL). The batch, along with the controls, were inoculated with 10 µL of *I. sakaiensis* culture (Abs ≈ 0.500 at OD600). A second series, using a 70% concentration of nutrient broth in vials, were inoculated, as the analytes were prepared in distilled water.

The cultures were incubated for 24, 48, 72, 96, and 120 h at 23 °C. Further, a continuous 72 h reading of the cultures at 23 °C was performed with the use of a plate reader (BMG Clariostar, Germany), with readings set at every 5 min after 200 rpm shaking for the two batches, as well as for a separate control series with 1% ethanol and one without ethanol (Appendix A). BPS was used as an analogue to BPA due to being the main replacement for BPA [25] and having a similar structure, but with a sulfur atom instead of a carbon atom. Estradiol was used to verify the probability of estrogenic substance affinity [26,27], as currently there are no reports on *I. sakaiensis* regarding this topic. For sterility, and to prevent analyte loss, the aqueous solutions of BPA and BPS were filtered through 0.2 µm PTFE syringe filters [28], while estradiol was not filtered due to high filter retention of this hormone [29], and a sterility control was performed in the case of all three stock solutions, by plating them onto nutrient agar, followed by a 72 h incubation at 23 °C.

### 2.4. Computational Simulation of BPA, BPS, and Estradiol as Substrate for I. sakaiensis PETase and MHETase

BPA, BPS, and estradiol were docked to *I. sakaiensis* poly(ethylene terephthalate) hydrolase (PETase) and mono(2-hydroxyethyl) terephthalate hydrolase (MHETase) structures that were identified in the Protein Data Bank under the codes 6EQH [30] and 6QGA [31], respectively. The structures of the ligands were retrieved from PubChem [32] (Figure 1).

Docking was performed using AutoDock4 and AutoDockTools4 [33]. All the ligands were treated as flexible; their torsion tree root and rotatable bonds are presented in Figure 1. The docking grid was centered in the active site of each enzyme, as identified from the literature: (i) residues Trp159, Ser160, Trp185, Asp206, and His237 in the case of PETase [30] and (ii) Ser225, Arg411, Ser416, Asp492, and His528 in the case of MHETase [31]. The same residues were considered flexible, while the rest of the protein was considered rigid. The genetic algorithm was used to generate 100 docking poses from which we analyzed the poses with the best estimates of binding free energies.

### 2.5. Growth Curve Statistical Analysis

ANOVA and *t*-tests were performed for the 120 h readings, whereas for the 72 h readings the growth curves were analyzed using the Advanced Data Analysis tool of ChatGPT [34,35]. Three fitting models were applied, namely the exponential model, the logistic model, and the Gompertz model [36]. The best model was selected based on the AIC values. Comparisons between the fitted curves and the individual parameters were performed using the ANOVA statistical test. The precondition for the normal distribution of the data was addressed with the Shapiro–Wilk test, considering as the null hypothesis that the data was drawn from a normal distribution [37]. The assumption of equal variances was tested using Levene’s test [38], with the null hypothesis of equal variances across the groups. The statistically significant differences were identified using Tukey’s HSD test [39]. The conversation with ChatGPT is provided in Appendix A.

## 3. Results and Discussion

### 3.1. Community-Level Physiological Profile and Strains Identification

Color development in the Biolog EcoPlate wells due to metabolic degradation provides a broad overview of the microbial community’s potential metabolic activity, serving as an indicator of the overall diversity in the bioactivity. The microbial community utilized 29 out of the 31 substrates contained by the microplate, with the greatest mean OD being that of carboxylic acids. It is plausible that within the original aquarium environment, the microorganisms developed the capacity to metabolize more types of carbon sources (Figure 2). The utilization of Tween 80 was notably higher in contrast to Tween 40; together with α-Cyclodextrin, the polymers were preferred after carbohydrate usage, with the exception of glycogen. Carboxylic acid was mainly preferred, with the exception of 4-hydroxy benzoic acid, which was less used by the microbial community, whereas 2-hydroxy benzoic acid was not consumed at all. The same pattern was observed in the case of the amines, with putrescine consumed at a low rate, and no consumption in the case of phenylethylamine.

Following the evaluation of the Biolog EcoPlates, species identification was performed by growing on media plates and inoculating individual colonies onto chromogenic agar, followed by the final identification via MALDI-TOF. The species were identified as belonging to: *Sphingomonas paucimobilis*, *Ralstonia pickettii*, *Massilia* sp., *Plesiomonas shigelloides*, and *Bacillus cereus*.

### 3.2. Evaluation of Species Growth during BPA Exposure

After the exposure to BPA of the identified species, there was the lack of a notable growth effect in the case of *Sphingomonas paucimobilis*, and little for the oligotrophic species *Ralstonia pickettii*, a slight inhibitory effect for *Massilia* sp., and minimal proliferation for *Plesiomonas shigelloides*. A more prominent growth effect was observed in the case of *Bacillus cereus* and *I. sakaiensis* (Figure 3). Due to the pathogenic nature of *B. cereus*, which could lower its value as a bioremediation agent, and to the fact it has already been reported in the literature as having the ability to metabolize BPA [40], further investigations were performed only on *I. sakaiensis.* The growth of *I. sakaiensis* in the presence of BPA was statistically significant, with a *p* < 0.05.

*I. sakaiensis* was grown in diluted media (70% nutrient broth) and 100% nutrient media in the presence of BPA, BPS, and estradiol, or in the absence of these compounds (Figure 4 and Figure 5, respectively). ANOVA and *t*-tests were performed for all conditions after each incubation time (Table 1).

In the case of the diluted media, significantly different results were obtained with the data after 72 h, 96 h, and 120 h of incubation. A *t*-test was applied for the results of each treatment, relative to the control conditions, after each growth interval. *I. sakaiensis* exhibited significantly increased growth with BPA, starting with the 72 h readings. The readings at 96 and 120 h also show that the growth of the bacteria treated with BPA significantly exceeds that in the control conditions (Figure 4). The growth of the bacteria in the presence of BPS and estradiol appears enhanced relative to the control, but the results are not statistically significant, except for the read at 72 h.

In the case of the 100% nutrient medium, the ANOVA tests show that significant differences were obtained after all incubation times, including 24 and 48 h. The significantly different results for the BPA, BPS, or estradiol relative to the control at each incubation time are labeled in Figure 5. BPA induces significant growth in *I. sakaiensis* relative to the control, starting with 48 h of incubation. The other compounds (BPS and estradiol), regardless of the nutrient percentage, did not induce a significant increase/decrease in the bacterial growth relative to the control, except for the results after 96 h.

### 3.3. Molecular Docking Results

The three ligands were docked to PETase and MHETase active sites [30,31] (Figure 6a,b). The estimated binding free energies for the best poses in each case are presented in Table 2. The predictions show that the ligands present an affinity with both targets, the most favorable interaction energies being obtained in the case of BPS, followed by estradiol and BPA. BPA and estradiol present lower binding energies to PETase, while BPS presents a lower binding energy to MHETase.

The interactions of the ligands with PETase and MHETase were visually inspected. We checked the superimposition of the current ligands with the ligands found in the crystal structures of PETase (2-hydroxyethyl methyl terephthalate, abbreviated to HEMT, PDB id 5XH3 [41]) and MHETase (4-(2-hydroxyethylcarbamoyl)benzoic acid, abbreviated to MHETA, PDB id 6QGA [31]), as seen in Figure 6c,d. The 2D interaction maps of BPA, BPS, and estradiol with PETase and MHETase are represented in Figure 7.

In the case of PETase, docked estradiol best overlaps the position of the bound HEMT (Figure 6c). The common residues interacting with the ligands are Gly86, Tyr87, Thr88, Trp159, Ser160, Met161, Ile208, and His237 (Figure 7a,c,e). These residues are part of the active site, and were identified as interacting residues in previous molecular docking studies of the substrate to PETase [30,42]. Depending on the structure of the ligand, different types of intermolecular interactions are established with the lining residues. For instance, Ser160 presents van der Waals interactions with BPA or estradiol, while it forms a hydrogen bond with BPS. His237 forms a pi-cation interaction with BPA, a van der Waals interaction with BPS, or an alkyl interaction with estradiol. From the three ligands, BPS goes deeper into the active site cleft, one of its phenyl moieties interacting through van der Waals interactions with Trp185, and through pi-alkyl interactions with Ala183. Overall, the results of the molecular docking show that the three ligands bind into the PETase active site, which could make them possible substrates of the enzyme.

In the case of MHETase, a phenyl moiety of BPS overlaps the aromatic cycle of bound MHETA, while BPA partially overlaps the tail of MHETA. Estradiol docks outside the active site (Figure 6d). The 2D interaction maps in Figure 7b,d,f show that the only residues common in the interactions by MHETase with BPA, BPS, and estradiol are Ser131, Ser225, His528, and Cys529. This small number of residues, considering the binding sites involving many residues, is due to the fact that estradiol binds outside the active site. When comparing the binding sites of BPA and BPS, we obtain the additional common residues: Gly132, Glu226, Phe495, Trp397, and Phe415. Some of these residues are mentioned in [31] as interacting with MHETA, a nonhydrolyzable MHET analog. These results show that BPA and BPS are possible substrates of MHETase. Even if estradiol presents a favorable binding energy to MHETase, not binding in the active site raises the question of whether it can be used as a substrate.

### 3.4. Growth Curve Analysis

The model that best describes the growth curves is the Gompertz model that leads to the smallest AIC value (~−6000), meaning that it best describes the growth dynamics across the curve, as well as the exponential and stationary phases. The logistic model led to a larger AIC value (~−5500).

The Gompertz model has the equation:(1)$$N\left(t\right)=A\exp\{-exp[\frac{um\;e}{A}\left(l-t\right)+1]\},$$
where *A* is the upper asymptote, *um* is the maximum growth rate, *e* is Euler’s number, and *l* is the lag time [36]. In Figure 8 we present a control growth curve fitted with the Gompertz model. The parameters of the growth curves recorded under all the conditions were analyzed using descriptive statistics, with the results presented in Table 3. As can be seen, *A* values are close to 1 for most conditions, which suggests that a similar maximum optical density was reached by the cultures across most conditions. The values of *um* present a small standard deviation, suggesting that the growth rate is consistent across the conditions. The lag time presents considerable variations (a large standard deviation), which suggests that the growth conditions explored here influence the time of entering into the logarithmic growth phase. BPA-induced metabolic changes could be responsible.

We compared the maximum growth rates (*um*) calculated from the growth curves. The normal distribution of *um* values across the conditions was proven by the Shapiro–Wilk test. The results of the ANOVA were an F-value of 1.03 and a *p*-value of 0.40. The large *p*-value shows that there are no statistical differences between the maximum growth rates across the different experimental conditions.

The time needed to reach half maximum (t_50) was calculated for the conditions and the results were compared to identify the statistical differences between the conditions. Prior to performing the ANOVA, the preconditions for the normal distribution of the data and of equal variances were tested. The Shapiro–Wilk test showed no violation of the normality distribution, while Levene’s test for equal variances showed that the variances are not equal across the groups (see Appendix A for details on null hypothesis and the resulting *p*-values for these tests). The ANOVA statistic F-value was 6.08 and the *p*-value was 2.12 × 10^−5^. The test thus proves that there are statistically significant variations in the t_50 values among the different conditions.

## 4. Conclusions

*I. sakaiensis* presented an increase in growth with BPA exposure compared to that reported for *B. cereus*. The growth in the 70% media concentration is significantly delayed relative to the undiluted nutrient broth conditions, with the metabolic efficiency being BPA > BPS > estradiol, but the differences are not always statistically significant. In this diluted medium, statistically different results were obtained for the data after 72 h, 96 h, and 120 h of incubation. Starting with 72 h, the growth of *I. sakaiensis* exposed to BPA exceeds that in the control conditions. A similar effect of BPA on *I. sakaiensis* growth is seen sooner, after 48 h of incubation in the 100% nutrient-rich medium. The other compounds, regardless of the nutrient percentage, did not induce a significant increase/decrease in growth relative to the control, except for results at 72 h in the case of the 70% nutrient broth medium, or after 96 h in the case of the 100% nutrient broth medium.

The present preliminary findings can have implications for bioremediation processes, as BPA is considered a persistent endocrine disruptor and, furthermore, *I. sakaiensis* is not known to have pathogenic characteristics, which can lower its value as a bioremediation agent. Most of the current reports on *I. sakaiensis* focused on PET degradation, however, PET does not contain BPA. Further studies are required as the usage of BPA as a carbon source is a new metabolic competence of *I. sakaiensis*, herein reported for the first time.

## Figures and Tables

**Figure 1 microorganisms-11-02891-f001:**
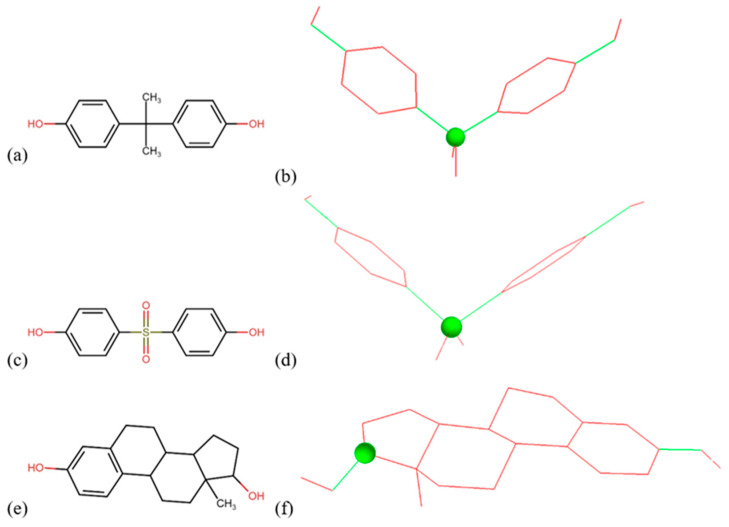
The 2D structures of BPA (**a**), BPS (**c**), and estradiol (**e**). The 3D structures of BPA (**b**), BPS (**d**), and estradiol (**e**), colored according to the rotatability of the bonds, using red for unrotatable and green for rotatable bonds. The roots of the torsion trees (**b**,**d**,**f**) are represented as a green sphere.

**Figure 2 microorganisms-11-02891-f002:**
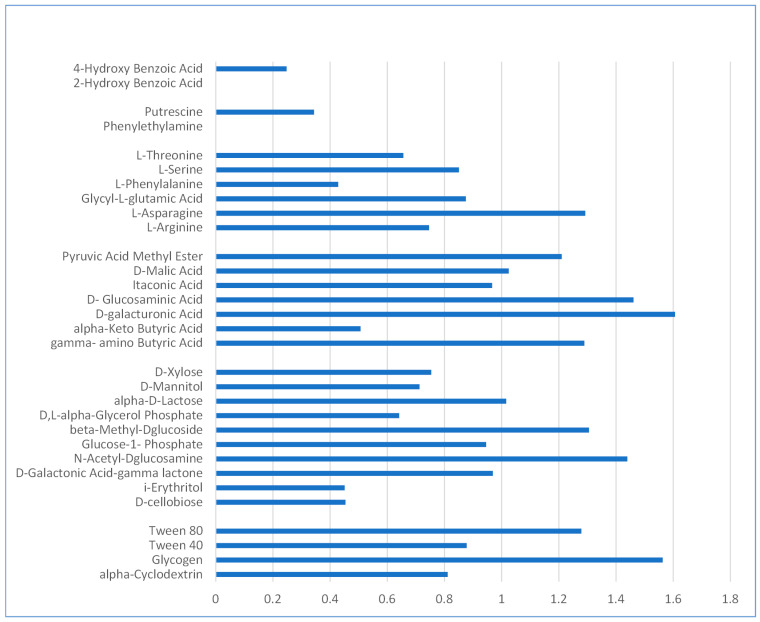
Substrate usage by the bacterial community (average over three replicates based on blank corrected, standard deviation is negligible).

**Figure 3 microorganisms-11-02891-f003:**
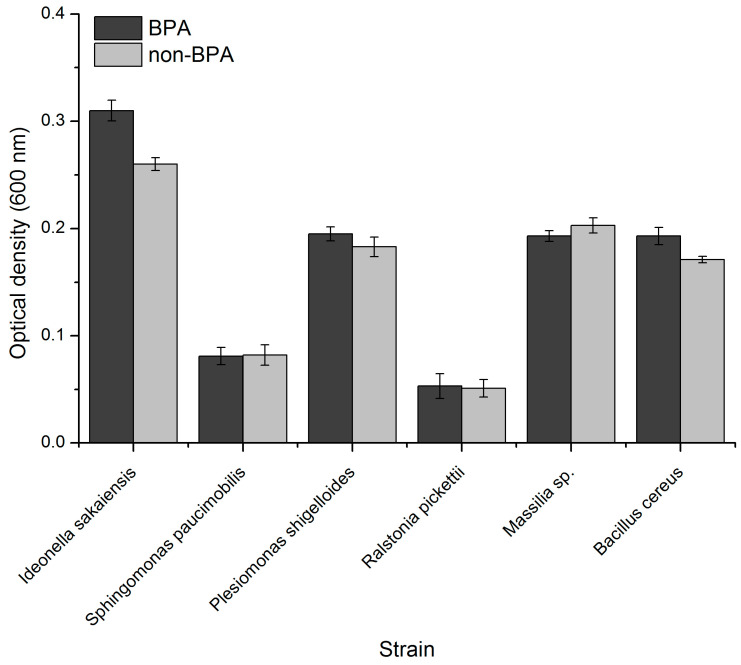
Identified species and their growth under BPA exposure.

**Figure 4 microorganisms-11-02891-f004:**
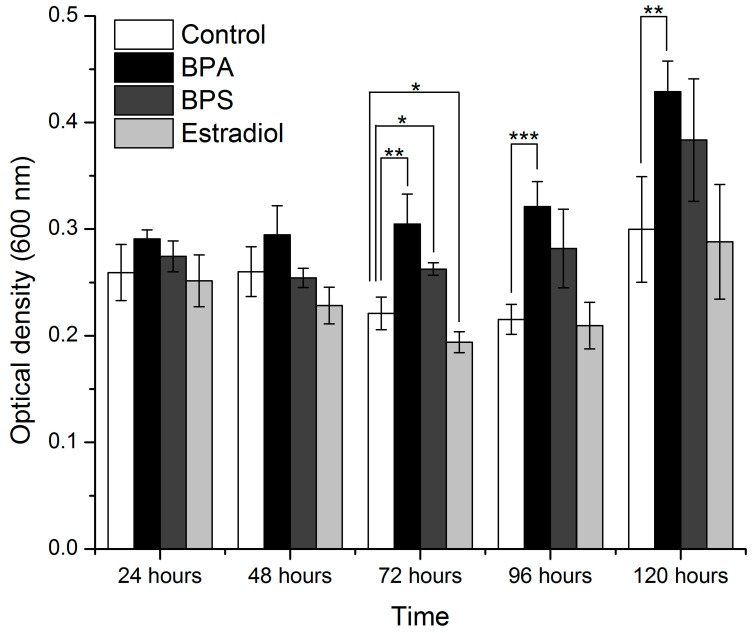
Mean optical density of *I. sakaiensis* (n = 4) measured under control conditions and under the treatment with BPA, BPS, and estradiol after 24, 48, 72, 96, and 120 h of incubation (70% nutrient media). The values that are significantly different relative to the values of the control condition at the same interval are marked with * for *p* < 0.05, ** for *p* < 0.01, and *** for *p* < 0.001.

**Figure 5 microorganisms-11-02891-f005:**
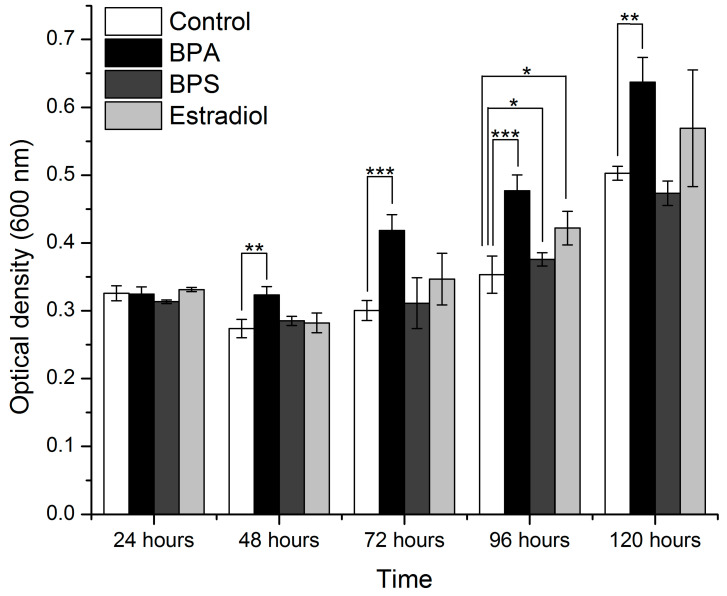
Mean optical density of *I. sakaiensis* (n = 4) measured under control conditions and under the treatment with BPA, BPS, and estradiol after 24, 48, 72, 96, and 120 h of incubation (growth medium with 100% nutrient broth). The values that are significantly different relative to the values of the control condition at the same interval are marked with * for *p* < 0.05, ** for *p* < 0.01, and *** for *p* < 0.001.

**Figure 6 microorganisms-11-02891-f006:**
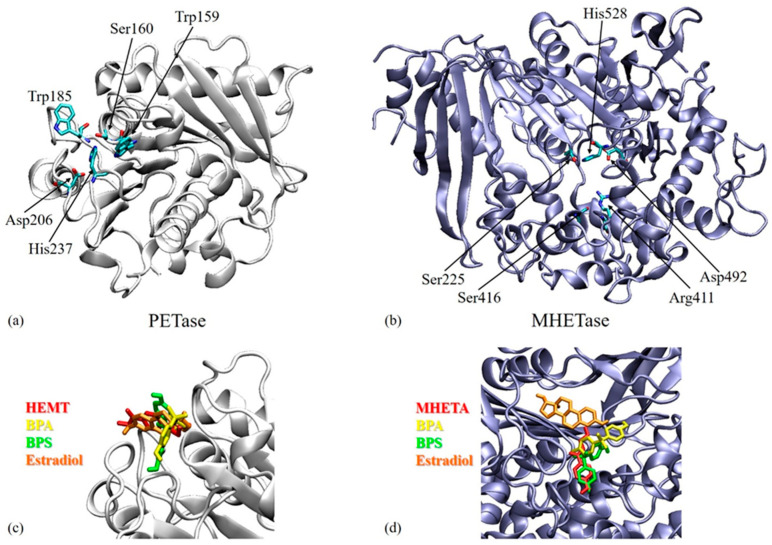
(**a**,**b**) The structures of PETase—PDB code 6EQH 1 (**a**) and MHETase—PDB code 6QGA 2 (**b**). The residues in the active sites of these models are represented and labeled on the figure. (**c**) Detail of the active site of PETase with the superimposed ligands: HEMT (2-hydroxyethyl methyl terephthalate), BPA, BPS, and estradiol. (**d**) Detail of the active site of MHETase with the superimposed ligands: MHETA (4-(2-hydroxyethylcarbamoyl) benzoic acid), BPA, BPS, and estradiol.

**Figure 7 microorganisms-11-02891-f007:**
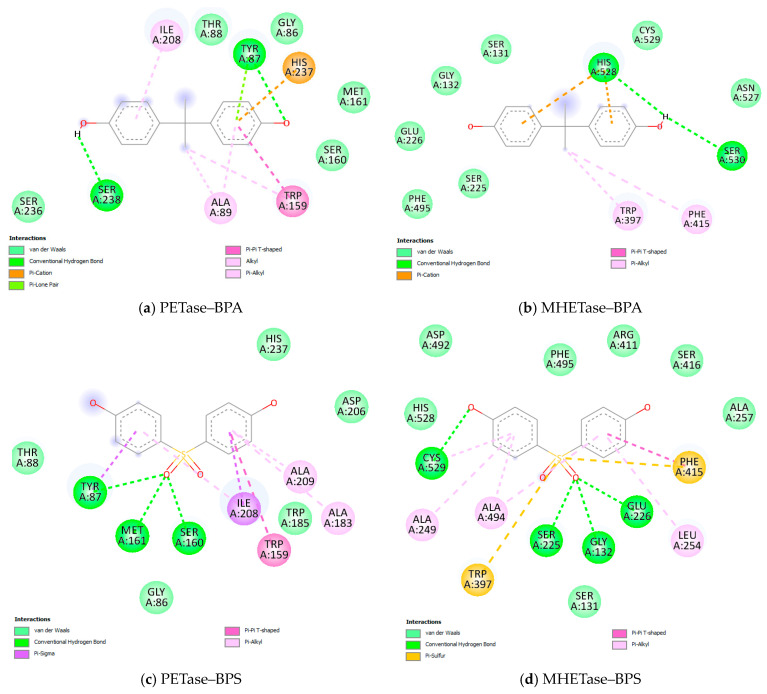
2D interaction maps of ligands with the two enzymes. (**a**,**c**,**e**) Show the interactions by BPA, BPS, and estradiol with PETase. (**b**,**d**,**f**) Show the interactions by BPA, BPS, and estradiol with MHETase. Interacting residues are labeled in circles and the interactions are presented as dashed lines colored according to interaction type, as presented in the legends. The maps were computed using Discovery Studio Visualizer v21.1.0.20298.

**Figure 8 microorganisms-11-02891-f008:**
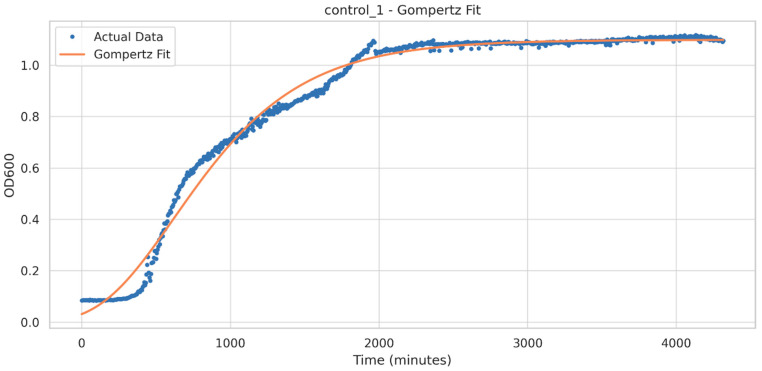
The control growth curve of *I. sakaiensis* plate readings during 72 h (blue dots), fitted with the Gompertz model (orange line).

**Table 1 microorganisms-11-02891-t001:** ANOVA test results obtained for the four conditions at each incubation time, when cells were grown in the medium with 70% nutrient or with 100% nutrient broth.

Incubation Time (Hours)	Medium with 70% Nutrient		Medium with 100% Nutrient	
	F-Value	*p*-Value	F-Value	*p*-Value
24	3.06815	0.06898	3.589544688	0.046460931 (significant)
48	7.19515	0.00508	13.13801863	0.000423948 (significant)
72	32.6886	4.68937 × 10^−6^ (significant)	12.55425	0.00052 (significant)
96	18.0501	9.60318 × 10^−5^ (significant)	23.77882	2.45 × 10^−5^ (significant)
120	7.72841	0.00388 (significant)	9.304546	0.001865 (significant)

**Table 2 microorganisms-11-02891-t002:** Estimated free energy of binding and inhibition constants (298.15 K) determined for BPA, BPS, and estradiol molecules docked at PETase and MHETase.

Target	Ligand	Estimated Free Energy of Binding (kcal/mol)	Estimated Inhibition Constant, Ki
PETase	BPA	−6.51	17.03 µM
	BPS	−7.68	2.33 µM
	Estradiol	−6.89	8.87 µM
MHETase	BPA	−5.87	49.53 µM
	BPS	−8.83	335.38 nM
	Estradiol	−6.64	13.61 µM

**Table 3 microorganisms-11-02891-t003:** Statistical analysis of the parameters determined for the Gompertz models that fit *I. sakaiensis* growth curves under all considered conditions.

	Upper Asymptote (A)	Maximum Growth Rate (um)	Lag Time (L)
Mean	1.0125	0.0008	120.3 min
Standard deviation	0.0971	0.0001	32.7 min
Minimum	0.8027	0.0006	1.4 min
25th percentile	0.9515	0.0008	110.6 min
Median	1.0356	0.0008	120.6 min
75th percentile	1.0808	0.0008	136.8 min
Maximum	1.1689	0.0010	176.6 min

## Data Availability

The data presented in this study are available on reasonable request from the corresponding author.

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
