# Peer review of "Ideonella sakaiensis* Can Metabolize Bisphenol A as a Carbon Source"

_microorganisms, 2023, doi:10.3390/microorganisms11122891_

Round 1
Reviewer 1 Report
Comments and Suggestions for Authors
This work studied the degradation of Bisphenol A by Ideonella sakaiensis, which expanded the knowledge of the metabolic competence and substrate spectra of I. sakaiensis. This work is meaningful for the biodegradation of bisphenol A and the application of I. sakaiensis in environmental bioremediation. However, some revisions are required to make the article more convincing.
1.Abstract, some key data should be included.
2.Line 15-18, “Following the growing need for non-pathogenic biodegradation microbial agents as an environmentally friendly and cost-effective solutions to eliminate such pollutants, this study aimed to investigate the degradation of Bisphenol A by Ideonella sakaiensis based on its (to date) unique enzymatic machinery that is already well known for degrading polyethylene terephtalate”.The statements before and after the comma lack a logical connection.
3.There is no mention of the experiment design of this research at the end of introduction.
4.Why 10 µg/mL was chosen as the experimental concentration, any reference?
5.Why choose Bisphenol S as an analog of Bisphenol A? Why BPS and estradiol were chosen as to compare with BPA treatments? Please explain.
6.There is no error bar in Figure 2 and Figure 3. Statistic analysis should be performed, and the variation were significant or not?
7.Figure 3-5, the lines in the background of the figures are suggested to remove, and those figures should be optimized to meet the publish requirement of the journal.
8.Line 180-181, “Strains identification was performed by innoculating individual collonies on chromogenic agar followed by the final identification via MALDI-TOF. ”; Line 175-179,“The Biolog EcoPlate method does present certain constraints, however, utilizing this method for assessing bacterial CLPP proves to be a more straightforward and cost effective option compared to DNA or RNA-based profiling. Furthermore, several investigations have indicated that molecular profiling surveys do not exhibit heightened sensitivity when compared to CLPP methods.”.Why is it that in Results and Discussion there are presentations and comparisons of identification methods, but even less analysis in Figure2?
9.In “3.2. Strains Growth Evaluation Under BPA Exposure” , there are some graphs and charts, but only a brief description of the graphical data and not the author's analysis.
10.The conclusion section makes no mention of the practical implications of this experiment.
Comments on the Quality of English LanguageSome parts of the article are not expressed accurately enough and could be improved.
Reviewer 2 Report
Comments and Suggestions for Authors
This manuscript is of substantial interest, sufficiently original, and the obtained results are important in the field of bioremediation. I recommend manuscript for publication after addressing the following.
It would be helpful if the authors add more information about possible sources of bisphenol A to the Introduction section. Bisphenol A is used as a hardener in the plastics industry and is a key monomer in the production of epoxy resins. It is also used in the synthesis of polysulfones and polyetherketones, as an antioxidant in some plasticizers and as an inhibitor of the polymerization of polyvinyl chloride.
Round 2
Reviewer 1 Report
Comments and Suggestions for Authors
The authors have revised the manuscript to address the issues raised by the reviewers and to improve the quality of the manuscript. It is now acceptable.
Comments on the Quality of English LanguageThe quality of the English language has improved compared to the first draft, and attention should be taken to grasp the logic of the English language to make the essay more reasonable.